# Best Practices for Measuring the Modulation Transfer Function of Video Endoscopes

**DOI:** 10.3390/s24155075

**Published:** 2024-08-05

**Authors:** Quanzeng Wang, Chinh Tran, Peter Burns, Nader M. Namazi

**Affiliations:** 1Center for Devices and Radiological Health, Food and Drug Administration, Silver Spring, MD 20993, USA; tranc@cua.edu; 2Department of Electrical Engineering and Computer Science, The Catholic University of America, Washington, DC 20064, USA; namazi@cua.edu; 3Burns Digital Imaging LLC, Rochester, NY 14450, USA; peter.david.burns@gmail.com

**Keywords:** endoscope, resolution, modular transfer function, spatial frequency response, ISO 8600-5, slanted edge, ISO 12233

## Abstract

Endoscopes are crucial for assisting in surgery and disease diagnosis, including the early detection of cancer. The effective use of endoscopes relies on their optical performance, which can be characterized with a series of metrics such as resolution, vital for revealing anatomical details. The modulation transfer function (MTF) is a key metric for evaluating endoscope resolution. However, the 2020 version of the ISO 8600-5 standard, while introducing an endoscope MTF measurement method, lacks empirical validation and excludes opto-electronic video endoscopes, the largest family of endoscopes. Measuring the MTF of video endoscopes requires tailored standards that address their unique characteristics. This paper aims to expand the scope of ISO 8600-5:2020 to include video endoscopes, by optimizing the MTF test method and addressing parameters affecting measurement accuracy. We studied the effects of intensity and uniformity of image luminance, chart modulation compensation, linearity of image digital values, auto gain control, image enhancement, image compression and the region of interest dimensions on images of slanted-edge test charts, and thus the MTF based on these images. By analyzing these effects, we provided recommendations for setting and controlling these factors to obtain accurate MTF curves. Our goal is to enhance the standard’s relevance and effectiveness for measuring the MTF of a broader range of endoscopic devices, with potential applications in the MTF measurement of other digital imaging devices.

## 1. Introduction

Endoscopes play a pivotal role in clinics and hospitals for assisting in surgery and disease diagnosis, including early detection and treatment of cancers, thereby enhancing the quality of patient care [1,2]. The millions of monthly endoscopic procedures conducted in the US, primarily for early cancer detection, are driving significant advancements in endoscopy. The effective use of endoscopes hinges on their performance, which can be assessed through a set of performance metrics such as resolution, geometric distortion, field of view (FOV), image intensity uniformity, and noise. Among these metrics, resolution stands out as a critical performance metric for endoscopes in clinical practice. High resolution allows clinicians to discern fine anatomical details, enabling the accurate identification of small structures, lesions, and pathological changes. This capability is crucial for the early detection of diseases, which often present as minute abnormalities. Enhanced resolution aids in distinguishing between different tissue types, essential for procedures like guiding devices and biopsies, where accurately targeting tissue significantly impacts diagnostic outcomes. High-quality images are also valuable in medical training, education, documentation, and communication (e.g., telemedicine consultations). Modern diagnostic tools and software, such as image analysis algorithms and artificial intelligence, rely on high-resolution images to function effectively. In summary, the resolution of endoscopes directly impacts their effectiveness in clinical practice by enhancing diagnostic accuracy, procedural guidance, and overall patient care. Therefore, the accurate assessment of resolution is paramount in ensuring the reliability and efficacy of endoscopic procedures.

### 1.1. Evaluation Metrics and Related Standards for Resolution

The contrast transfer function (CTF) and modulation transfer function (MTF) serve as comprehensive metrics for evaluating endoscope resolution. The CTF and MTF measure contrast (C) and modulation (M) loss, respectively, as a function of spatial frequency in endoscope images [3]. If images are not artificially processed, both C and M generally decrease with spatial frequency, reaching a point where details are no longer visually resolved.

The CTF is assessed using a test chart of square-wave patterns (e.g., bars) with varying spatial frequencies/sizes, while the MTF can be measured with a test chart of sine-wave patterns. According to Fourier theory, a square wave can be represented as the sum of a series of sine-wave components, with frequencies that are odd multiples of the fundamental frequency (harmonic frequency) [4]. Consequently, a square wave CTF can be quite accurately converted to its equivalent sine wave MTF using a series expansion formula, or vice versa [5]. If an image is not artificially processed, the CTF is often higher at all frequencies than the corresponding MTF. The formula for conversion between CTF and MTF can be simplified by only considering the first term of the series expansion (i.e., the fundamental frequency term), as MTF≈(π/4) CTF for high frequencies, especially for frequencies higher than one-third of the cut-off frequency [6,7]. The theory of the MTF is more straightforward compared to that of the CTF and is extensively employed for evaluating optical system performance.

A more profound comprehension of the MTF should commence with the optical transfer function (OTF). The OTF is a function that characterizes the ability of an optical system to capture or transfer the modulation and spatial phase of a test chart, with spatial frequency as the independent variable. It is a complex function whose modulus is the MTF and whose argument is the Phase Transfer Function (PTF). Both the MTF and PTF can be derived from the OTF. The PTF holds significance in certain applications, such as interferometry or situations where maintaining phase information is critical. The overall OTF of a linear system is equal to the product of the OTFs associated with several individual components (e.g., lenses, sensors). For many measurements, the PTF is less important, as in the incoherent illumination of a target and detection of an image by a sensor, for example. Similarly, the overall MTF of a linear system is equal to the product of the MTFs associated with its individual components.

For most optical systems, the MTF is often prioritized for several reasons. Firstly, the MTF is often more directly correlated with the perceived image quality (e.g., image sharpness and contrast) by human observers [8,9]. Human vision, being sensitive to changes in contrast and resolution, makes the MTF a perceptually meaningful metric. Secondly, the MTF provides a simpler interpretation compared to the OTF, offering a clear indication of how well an imaging system can capture fine details. This simplicity is crucial in various applications, including photography and medical imaging. Thirdly, the MTF is more directly applicable in practical scenarios, such as designing camera systems or evaluating lens performance. It enables engineers to comprehend how the system responds to information at different spatial frequencies, aiding in the optimization of designs for specific applications. Lastly, the MTF finds frequent use in image processing and computational photography, playing a crucial role in algorithms aimed at enhancing image sharpness, correcting optical aberrations, or improving overall image quality.

The MTF has become a standard metric for characterizing the optical performance of imaging systems, and several international standards on OTF or MTF measurement have been developed. These standards aim to facilitate the communication and comparison of the capabilities of different devices among manufacturers, researchers, and users. While there may be some overlaps, each standard emphasizes specific aspects. The ISO 9334 standard establishes terminology for OTF measurement [10], providing a foundation for consistent terminology use. Meanwhile, the ISO 9335 standard delineates the principles and procedures governing the measurement of the OTF [11], offering a standardized approach to ensure consistency in measurement practices. Additionally, the ISO 9336 standards address specific applications of the OTF across diverse optical and electro-optical fields, with each standard dedicated to a particular application [12,13,14]. The ISO 11421 standard goes further by providing comprehensive guidance on the techniques and procedures for assessing the accuracy of OTF measuring equipment. It also outlines methods for estimating the uncertainty in measurements conducted on specific imaging systems, contributing to the reliability and precision of OTF measurements [15]. On the other hand, the ISO 15529 standard describes principles for the measurement of the MTF in sampled imaging systems [16]. It is essential to note that these standards are designed for general optical systems that have little geometric distortion and are used for long-distance imaging. On the other hand, endoscopes use small, high-resolution sensors optimized for close-up views and low-light conditions. They often have a short focal length and a wide FOV [17], resulting in severe geometric distortion [18]. Therefore, these standards cannot be directly applied to endoscope evaluation due to the unique characteristics and requirements of endoscopic devices [17,18]. Consequently, the evaluation of endoscopes requires tailored standards that address the specific intricacies of these instruments and their applications.

The ISO Technical Committee 172 Subcommittee 5 Working Group 6 (ISO/TC 172/SC 5/WG 6) holds the responsibility for overseeing international endoscope standards, operating mainly within the ISO 8600 standard series. Two versions of the ISO 8600-5 standards have been developed to determine the optical resolution of rigid endoscopes with optics. The 2005 version [19], which was withdrawn in 2020, introduced a simple limiting resolution metric. This metric is defined as the highest spatial frequency perceivable at a given working distance of the endoscope. It serves as a single-valued subjective metric suitable for rapid manufacturing testing, quality assurance (e.g., in end-of-line endoscope assembly testing), or for providing a straightforward metric easily understood by lay users [3]. However, this metric cannot comprehensively evaluate endoscope resolution, particularly for spatial frequencies lower than the limiting resolution. Even if an endoscope exhibits a high limiting resolution, it might still perform inadequately in imaging anatomical details with spatial frequencies lower than the specified limit. To address this limitation, the 2020 version of the ISO 8600-5 standard introduces two additional metrics—the CTF and MTF [3]. Different from the single-value subjective metric of limiting resolution, the CTF or MTF offers a detailed and objective assessment of image quality by measuring the camera’s ability to reproduce contrast or modulation at various spatial frequencies.

Nevertheless, the main drawback of the 2020 version of the ISO 8600-5 standard is its reliance on a theoretical understanding of MTF measurement, lacking empirical support from endoscope bench test data. This absence of practical validation raises concerns about the standard’s real-world applicability and accuracy in assessing endoscope performance. Another limitation is that the standard is exclusively applicable to rigid endoscopes with optics, excluding opto-electronic video endoscopes from its scope, which significantly impacts its applicability. Video endoscopes, which are increasingly common in modern medical procedures, have unique characteristics such as electronic image capture and digital processing that are not addressed by the standard. The exclusion undermines the standard’s ability to comprehensively cover the diverse range of endoscopic technologies in use today, potentially affecting clinical outcomes and the adoption of advanced digital endoscopic techniques.

### 1.2. Measurement as Estimation

While the basic principles of the OTF and MTF for linear systems are well understood, a distinction needs to be made regarding imaging performance measurement as an estimation task. We can consider MTF measurement as an estimate of a system performance metric crucial for evaluating the resolution (ability to capture spatial details) of an endoscopic system. A standard method, for any application, should provide a consistent estimate of the system performance metric. Efforts to develop ISO standard methods for imaging performance evaluation have largely focused on minimizing measurement error. Two critical statistical measures of errors are bias and variability.

A robust method should estimate the MTF while minimizing bias introduced by factors such image noise, optical distortion, and image processing steps unrelated to spatial detail. An example of the image processing steps is the application of the gamma curve, a look-up table that is part of all imaging chains. This operation is applied at the image level and ideally has minimal impact on image detail and, consequently, on the measured MTF. The influence of this step can be mitigated by adopting a moderate-contrast edge feature for analysis. 

### 1.3. Our Contributions

Our efforts documented here aim to evaluate the current standard method for MTF evaluation of video endoscopes and provide recommendations for its proper use and future revision. As we will demonstrate, various operational choices, such as image compression, can influence measurement results. Because these operations involve spatial processing, they affect system performance, and it is crucial for the measured MTF to reflect these influences. In cases like gamma correction or illumination non-uniformity (vignetting), we strive to minimize their impact on MTF results or propose testing methods that mitigate these effects.

Specifically, our goal is to broaden the scope of the ISO 8600-5:2020 standard [3] to encompass opto-electronic video endoscopes. This involves evaluating and optimizing the MTF test method outlined in the current standard. Additionally, we examine parameters that can impact MTF measurement accuracy but are not explicitly addressed in the standard. These include intensity and uniformity of image luminance, chart modulation compensation, linearity of image digital values, auto gain control (AGC), image enhancement, image compression, and the dimensions of the region of interest (ROI) for edge images. This comprehensive analysis aims to enhance the standard’s relevance and effectiveness in evaluating a wider range of endoscopic devices. To our knowledge, this study represents the first comprehensive effort to establish best practices for the MTF evaluation of video endoscopes.

## 2. Methodology

The ISO 12233 standard outlines methods for evaluating the spatial resolution capabilities of a digital camera, gaining popularity for precise MTF measurement, referred to as spatial frequency response (SFR) in this standard [20]. The standard establishes terminology and offers detailed guidelines for test conditions (e.g., illumination and device settings), test charts, methods, and software code for MTF calculation. The ISO 12233 standard refers to the results of the slanted-edge method as the edge-spatial frequency response (e-SFR). We can interpret this e-SFR as a measurement (or estimate) of the pre-sampled MTF based on the ISO 12233 method. In the same way, the sine-wave method provides a different estimate (s-SFR) of the MTF. The SFR, rather than the MTF, is used since digital still and video cameras are usually not strict linear systems due to various parameters (e.g., sampling, color-filter interpolation, and non-linear image processing). In this article, due to common usage in endoscopic imaging, we will refer to the e-SFR as the measured MTF. Notably, the ISO 8600-5:2020 standard [3] was developed, in part, based on the ISO 12233 standard. We conducted experimental MTF measurements primarily following these two standards.

### 2.1. Main Devices

#### 2.1.1. Endoscopic System

This study primarily used an Olympus EVIS EXERA II endoscopic system (Olympus America, Center Valley, PA, USA), referred to as “the endoscope” in this paper. The endoscope comprises a high-intensity xenon light source (CLV-180), a gastrointestinal videoscope (GIF-H180), and a video system center (CV-180). It offers various parameter settings to control light intensity and image capture. The xenon light source features adjustable intensity settings, capable of manual or automatic control. The AGC function can be toggled on or off, and multiple image enhancement options can be selected. Captured images can be saved in TIFF or JPEG format. Further details on these parameters will be discussed in the subsequent sections.

#### 2.1.2. Test Charts and Measuring Distance

The MTF can be measured using a sine-wave chart directly or through a point or line source to calculate it via the point or line spread function. Common line sources include slit [10,11,15,16] and edge [20] sources. Among these, edge charts are gaining popularity due to their simplicity in production and use.

The ISO 12233 standard recommends two types of charts for MTF measurement: a sine-wave-based chart and an edge-based chart. Each is associated with its method and algorithm for obtaining the s-SFR and e-SFR, respectively. Sine-wave-based charts consist of alternating bright and dark bands in a sinusoidal pattern, providing detailed frequency-specific MTF data by measuring contrast at various frequencies. Edge-based charts, featuring a sharp transition between dark and light areas, derive the MTF from the edge spread function and are simpler to set up and analyze. 

An essential prerequisite for MTF measurement is the isoplanatic behavior of the digital camera, meaning its point spread function is independent, within a specified accuracy, of the position of the conjugate point source in the object plane [10]. However, endoscopes, characterized by severe geometric distortions (i.e., variable local magnifications with image positions) [18] and aberration depending on field angle, do not exhibit isoplanatic behavior. Consequently, it becomes crucial to use a small ROI of the chart image for MTF calculation, so that the ROI can be considered locally isoplanatic. Therefore, a sine-wave-based chart is too large to meet the isoplanatic requirement. An edge-based chart is a better choice since only a small ROI of the edge image is needed to calculate the MTF.

In this research, we used an extended ISO 12233:2017 Edge SFR chart from Imatest LLC, referred to as “the chart”, for simplicity. The endoscope’s depth of field (DOF) spans from 2 to 100 mm. Since the endoscope’s prime lens has a fixed focal length, the MTF varies with measuring distance. A higher quality MTF measurement chart is required at a shorter distance due to the larger magnification. Shorter measuring distances amplify any imperfections along the edge, making it harder to achieve a smooth transition between the edge and background, which is a fundamental requirement for accurate MTF calculation. While the chart possesses proper edge contrast [20] and good diffusing scattering properties, its edge quality is not sufficient for MTF measurement at distances shorter than 80 mm (refer to Section 2.3.2 for details). Therefore, we chose 80 mm as the measuring distance. MTF measurements at shorter distances were excluded because the chart’s quality cannot ensure accurate MTF measurements. Our primary focus is on exploring various factors that influence MTF measurement and optimizing the measurement method for endoscopes’ MTF, rather than evaluating this endoscope’s overall performance at different distances.

When assessing the impact of linearity of image digital values on MTF measurement, both the chart and a back-illuminated high-contrast USAF 1951 chrome-on-glass chart (R3L3S1P, Thorlabs, Inc., Newton, NJ, USA) were employed to represent two scenarios: low-contrast chart edge and high-contrast chart edge. The chrome-on-glass target was exclusively utilized for this factor and was not employed in any other experiments.

#### 2.1.3. Light Sources

We used two light sources to examine the impact of the intensity and uniformity of image luminance on MTF measurements: the inherent xenon light source of the endoscope, referred to as xenon light, and a pair of incandescent lamps (EiKO 01960 Supreme Photoflood lamps, EiKO Global, LLC, Olathe, KS, USA), referred to as incandescent light for simplicity. The xenon light directly illuminated the chart from the front. The incandescent lamps were covered with a scattering cover, illuminating the chart from the front on both sides at a 45° angle [21], with lamp intensity regulated using a lamp dimmer. The illuminance on the chart by the incandescent lights was much more uniform than the xenon light.

The xenon light operated in the normal intensity mode, offering automatic or manual brightness control function. Automatic control maintained a relative stable illuminance on the chart by automatically adjusting the light luminance when the chart distance changed. In contrast, manual control maintained a consistent luminance level, but the illuminance on the chart decreased as the chart distance increased. Both light sources were stationary at a given distance, while the chart was moved to different locations.

When evaluating the effects of linearity of image digital values using the USAF 1951 chrome-on-glass chart, we back-illuminated it with a small LED array light. The LED light was covered with several layers of white scattering paper, placed in direct contact with the front surface of the LED light and the back surface of the glass chart to achieve uniform illumination.

#### 2.1.4. Illuminance Meter

We measured the illumination falling on the chart at locations that were then used for the imaging performance evaluation. The illuminance at various locations in the object space was measured using an illuminance spectrophotometer (CL-500A, Konica Minolta USA, Ramsey, NJ, USA), referred to as an illuminance meter for simplicity. To measure the illuminance at a specific location on the chart, the receptor window of the illuminance meter was placed at the same chart surface position facing the light source. The chart was temporarily moved during the measurement to provide space for the meter. The meter was used to measure the illuminance at different positions and under different brightness control modes (auto and manual) and brightness level settings at different distances. The measured data were further used to calculate the opto-electronic conversion function (OECF) [22] for gamma correction, as discussed in Section 2.3.3.

### 2.2. Experimental Method

#### 2.2.1. Setup

Two fiber optic positioners (FPR2-C1A, Newport Corporation, Irvine, CA, USA), affixed to height-adjustable posts, were employed to maintain the optical axis of the endoscope both horizontal and perpendicular to the chart surface. The test chart was affixed to a customized height-adjustable metal plate with magnets. This metal plate was vertically mounted on a translation stage (TBB1212, Thorlabs, Inc.) that can move in two rails along the endoscope’s optical axis. This setup (Figure 1) facilitated adjustments to the distance between the chart and the distal end of the endoscope. Figure 1 shows the experimental setup when using the inherent xenon light source. The ISO 12233: 2023 standard illustrated the test chart illumination method when using external light sources [20].

#### 2.2.2. ROIs on Chart Images

The endoscope’s MTF varies across different locations within the FOV. While measuring MTF at every location is impractical, defining a limited number of representative locations within the FOV helps facilitate a comparison of the results. The 8600-5:2020 standard mandates the measurement of the MTF at five specific locations within an endoscope image. These include the image center and points at 70% of the maximum image height and width from the center (points A, B_1_, B_2_, B_3_, and B_4_ in Figure 2). It is crucial to note that due to significant geometric distortion, a straight line on the test chart may appear curved in the endoscope image unless it passes through the image center, assuming circularly symmetric distortion [18]. To ensure a straight edge in the image, the chart should ideally be aligned with the edge being horizontal at B_2_ and B_4_, vertical at B_1_ and B_3_, and both horizontally and vertically at A, as illustrated by the gray bars in Figure 2. Consistent with the common tilted edge angle [20], the edges were oriented at approximately 5° from the horizontal or vertical axis. This intentional misalignment ensured that the edge being analyzed did not align precisely with the pixel sampling, resulting in the presence of multiple sampling phases and allowing for the reconstruction of an oversampled edge profile. This slight tilt angle did not cause the straight edge to become curved, although this is not a critical restriction for the most recent analysis method for the edge MTF [23].

Given that our study primarily focuses on method development rather than endoscope evaluation, we exclusively presented MTF data at locations A and B_2_. These two locations are deemed representative: Center location A is of utmost significance as it possesses the largest magnification [18], providing the best illustration and image quality. On the other hand, location B_2_ is positioned the farthest from A, characterized by inferior illumination, and smaller magnifications [18], serving as a scenario with the worst conditions within these five locations.

### 2.3. Factors under Examination

We studied a series of factors that might affect endoscope MTF measurement. When one factor was examined, the other factors were set at default values. All measurements were conducted in a dark environment as background radiation, especially specular reflections, can cause a rapid drop in the measured MTF as the spatial frequency increases from its nominally zero value [11]. There was no direct illumination of the camera lens by the light sources. The area surrounding the test chart had low reflectance to minimize flare light. The captured chart images were saved as full-height images with dimensions of 1280 × 1008 pixels on a CompactFlash card through a PC card adaptor connected to a USB port of the CV-180 and then transferred to a computer for MTF calculation.

For the MTF calculation, we used a modified version of the MATLAB (version R2022b) script *sfrmat3*. Note that the current ISO 12233:2023 standard includes several refinements from that implemented in *sfrmat3*. The current implementation is *sfrmat5* [24]. However, the conclusions drawn in this study are still valid.

A series of factors that might affect MTF measurement were studied. To ensure a consistent comparison, the default settings of these factors are listed in Table 1 if not specified. Details for these factors are explained in different subsections, as follows. The flowchart in Figure 3 shows the detailed procedures for how data were collected and analyzed step by step.

#### 2.3.1. Intensity and Uniformity of Image Luminance

In an imaging chain, the light illuminating a test chart transmits through or is reflected by the chart, then passes through the camera lens and reaches the camera sensor to generate digital images. Illuminance measures the total light incident on the chart’s surface, while chart luminance gauges the amount of light from the chart surface, influenced by both the illuminance and the transmittance or reflectance of the chart. The illuminance and the chart luminance can be mutually converted [22]. Pixel intensity in captured images, often in grayscale, is called image luminance. It is important to note that image luminance, tied to pixel intensity, differs from chart luminance, which is a measure of physical surface properties.

The intensity and uniformity of image luminance can have an impact on MTF measurement. Various factors, including the properties of the light source, the reflectance or transmittance of the test chart, and the optical characteristics of the camera lens, along with camera aperture and exposure time, can collectively influence the intensity and uniformity of image luminance.

In our study, we selected the endoscope and test charts, but the aperture and exposure time of the endoscope cannot be controlled. Therefore, we tried to understand the effects of the intensity and uniformity of image luminance on MTF measurement by manipulating the intensity and uniformity of the illumination light using xenon and incandescent light sources. The xenon light is often used in real-world endoscopic applications, while the incandescent light served as a reference. The incandescent light was much more spatially uniform than the xenon light. To assess the uniformity of image luminance, we captured images of a 10-inch-by-10-inch diffuse reflectance target with 50% reflectance (Spectralon^®^ SRT-50-100, Labsphere Inc., North Sutton, NH, USA). White balance was performed before image capture. The target covered the whole FOV.

The intensity of the light source is directly related to the illuminance on the test chart, which in turn affects the intensity of chart luminance and image luminance. Therefore, we evaluated the effect of image luminance intensity on the MTF by varying the light source intensity. The xenon light was controlled through the endoscope control panel, and the incandescent light was controlled through a dimmer. The MTF curves at different intensity levels were calculated based on the chart images captured at 80 mm. Additionally, we employed the automatic control of the xenon light for MTF measurement. It is important to note that smaller level numbers for the same light indicate lower intensity, but they do not have a quantitative meaning. Furthermore, the levels for the two light sources are not directly comparable.

#### 2.3.2. Chart Modulation ($M_{chart}$) Compensation

In theory, the directly measured MTF ($M_{measured}$) based on the test chart is the product of the endoscope MTF and the chart modulation ($M_{chart}$). Therefore, $M_{measured}$ should be compensated by $M_{chart}$ to obtain the endoscope MTF, as $MTF=M_{measured}/M_{chart}$. If $M_{chart}$ is not high enough, there will be a noticeable difference between $M_{measured}$ and the actual endoscope MTF: $M_{measured}$ will be lower than the endoscope MTF. In that case, the endoscope MTF will be underestimated if we used $M_{measured}$ directly as the endoscope MTF, i.e., bias is introduced into the measurement.

In our study, $M_{chart}$ was provided by the manufacturer as $M_{chart}=\exp\left(-0.0200f_{chart}-{\left(0.2456f_{chart}\right)}^{2}\right)$, where $f_{chart}$ is the spatial frequency in cycles per mm in the object space (cy/object mm). To evaluate the impact of $M_{chart}$ compensation on the endoscope MTF, we calculated and compared the difference between $MTF_{measured}$ and the MTF based on our measurements. Additionally, we artificially generated different levels of $MTF_{measured}$ and $M_{chart}$ to further examine the impacts of $M_{chart}$ compensation. Please notice that the MTF mentioned in this paper refers to the endoscope MTF after $M_{chart}$ compensation, unless specified otherwise.

#### 2.3.3. Linearity of Image Digital Values

Linearity, like isoplanarity, is a fundamental prerequisite for accurate MTF measurement, requiring a proportional response by an imaging system to the level of the input signal. However, in most digital images, pixel values ($V_{encoded}$) are not linear to scene luminance ($V_{original}$). Cameras typically convert $V_{original}$ to $V_{encoded}$ through encoding gamma correction, or pre-compensation, with equation $V_{encoded}=V_{original}^{\gamma }$, where the gamma value (γ) that is usually less than one, and both $V_{original}$ and $V_{encoded}$ are normalized with the intensity just saturating the sensor as one. This correction mimics the non-linear perception of brightness change by the human eye, enhancing image quality by preventing issues like banding and loss of detail, especially in shadow regions. Before displaying, decoding gamma correction, or post-compensation, is applied through $V_{display}=V_{encoded}^{1/\gamma }$, ensuring consistent perceived brightness with the original scene. $V_{display}$ is mathematically equal to $V_{original}$. This correction also aids in maintaining image consistency across different devices. Overall, the encoding and decoding gamma correction steps enhance the linearity of the imaging chain, improving the accuracy of brightness representation in the final displayed images.

The chart images after encoding gamma correction ($V_{encoded}$) should be linearized through decoding gamma correction before MTF calculation [20]. The gamma value for our endoscope was estimated by measuring the opto-electronic conversion function (OECF) [22] using the 20 gray patches (Figure 4) on the chart. Due to vignetting, different spots with the same luminance in the scene might have different $V_{original}$ values, thus affecting the γ estimation accuracy. To diminish the vignetting impact, it would be best to position the chart at a relatively long distance so that the patches are captured near the center of the FOV, where the vignetting is minimal, or align the patches symmetrically to the optical center, assuming the vignetting is also symmetric. The visual density values ($D_{i}$ where i is patch number) of each patch are listed in Table A.3 of ISO 14524 [22]. The illuminance on each patch ($E_{i}$) was measured with the illuminance meter and the luminance of each patch ($L_{i}$) was calculated as $L_{i}=10^{-D_{i}}E_{i}/\pi$. An OECF curve illustrates image pixel values as a function of chart luminance. Using the trendline equation of this curve, the maximum luminance ($L_{max}$) to just saturate the image sensor (i.e., pixel value 255 for an 8-bit image) was derived. The pixel values were normalized by 255 to achieve $V_{encoded}$ and the luminance values were normalized by $L_{max}$ to achieve $V_{original}$. Based on the equation $V_{encoded}=V_{original}^{\gamma }$, γ was derived. The measured γ for our endoscope is 0.45, the same as for most imaging systems. The equation $V_{display}=V_{encoded}^{1/\gamma }$ was used for decoding gamma correction to linearize the edge images before MTF calculations. The linear values of $V_{original}$ and $V_{display}$ are theoretically identical.

To evaluate the impact of signal linearity on MTF calculations, MTF curves based on three popular encoding γ values of 0.36, 0.45, and 0.56 without decoding gamma correction were compared with linear data (i.e., γ = 1). The captured edge images (encoded with γ = 0.45) were decoded as $V_{\gamma =1}=V_{encoded}^{1/0.45}$ to obtain linear data (understood as $V_{original}$ being encoded with γ = 1). The images with encoding γ of x can be obtained by $V_{\gamma =x}=V_{\gamma =1}^{x}$. It should be noted that the exact encoding and decoding equations for gamma correction might vary slightly depending on the image encoding methods (e.g., sRGB, Adobe RGB). Additionally, $V_{original}$, $V_{encoded}$, and $V_{display}$ in this section are all normalized. They were converted to 8-bit image values by multiplying 255 for MTF calculation.

#### 2.3.4. Auto Gain Control (AGC)

AGC dynamically adjusts the gain (amplification applied to the incoming signal) to ensure a balanced overall image brightness within an acceptable range, particularly in changing lighting conditions. In this study, we captured chart images by enabling and disabling the AGC function of the endoscope, and then calculated and compared the MTF curves based on these images.

#### 2.3.5. Image Enhancement

The endoscope is equipped with an image enhancement function designed to electrically enhance the sharpness of captured images. This function offers three enhancement modes: structural enhancement A, which enhances the contrast of fine patterns in images; structural enhancement B, which enhances the contrast of even finer patterns than A in images; and edge enhancement E, which specifically enhances the edges of images. Each enhancement mode includes eight levels (A1 to A8, B1 to B8, and E1 to E8), with level 1 having the least enhancement and level 8 having the strongest, resulting in a total of 24 possible enhancement approaches. For our study, we only selected A1, A8, and B1 to illustrate the impact of image enhancement on MTF measurement.

#### 2.3.6. Image Compression

Images from endoscopes are usually compressed through either lossless or lossy methods and saved in various image file formats. Among the most popular formats are the lossless Tagged Image File Format (TIFF) and lossy Joint Photographic Experts Group (JPEG). Typically, TIFFs can be uncompressed or lossless compressed and preserve all original image data, thereby achieving the highest image quality, but this comes with the cost of a larger file size compared to JPEG. We assume the TIFF images used in this research were uncompressed, which will not affect the results and conclusion. On the contrary, JPEG uses lossy compression, potentially discarding certain image information to reduce file sizes. In JPEG, different compression quality levels are available, balancing file size and image quality. Our endoscope can save images as TIFF files or three different qualities of JPEG files: super high quality (SHQ), high quality (HQ), and standard quality (SQ). The MTF curves calculated from these image file formats were compared.

To further investigate the impact of image compression, two high-quality images were compressed with MATLAB. One was an endoscope TIFF image captured in our lab, and the other was a high-quality edge image from an online source. The images were compressed to different qualities of JPEG images (MATLAB codes to compress ‘test.tif’ to ‘test_XX.jpg’: *im = imread(‘test.tif’); imwrite(im, ‘test_XX.jpg’, ‘Quality’,XX);*). The ‘XX’ in the code is a quality scalar in the range [0, 100]. A higher quality scalar indicates better image quality but results in larger file sizes. Quality scalars between 90 and 100 are considered high quality, 70–89 medium quality, and 1–69 low quality. MTF curves were compared for JPEG quality scalars of 20, 40, 60, 80, and 100.

#### 2.3.7. ROI Dimensions

While the center locations of the ROIs have been defined in Figure 2, it is important to understand the potential impact of their dimensions on the calculated MTF. The MTF is derived from the edge spread function, which relies on the binning of rows/columns perpendicular to the edges within the selected ROIs. Larger ROIs may yield a more smoothed edge spread function and subsequently a smoother MTF profile. However, as explained in Section 2.1.2, achieving isoplanarity is also important for accurate MTF measurements. The point spread function of an endoscope is contingent upon the position of the conjugate point source within the object plane, suggesting that endoscopic systems do not adhere strictly to isoplanatic behavior due to significant geometric distortions and aberrations [18]. Therefore, using smaller ROIs could approximate the isoplanarity requirement within the ROI (i.e., locally isoplanatic).

We calculated MTF curves based on different ROI dimensions in pixels, both parallel and perpendicular to the edge direction, with the centers for various ROI dimensions kept the same as defined in Figure 2. The curves were compared to evaluate the impacts of ROI dimensions on MTF calculation.

## 3. Results

We have evaluated the impacts of several factors on MTF measurements, including the intensity and uniformity of image luminance, $M_{chart}$ compensation, the linearity of image digital values, AGC, image enhancement, image compression, and ROI dimensions. The results are discussed as follows. While conducting MTF calculations, the captured color RGB images were converted to grayscale images. This conversion was achieved using the coefficients of (0.213, 0.715, 0.072) for the red, green, and blue channels, respectively, as defined by ITU-R BT.709 [25].

### 3.1. Intensity and Uniformity of Image Luminance

The intensity and uniformity of image luminance are directly influenced by the intensity and uniformity of the illumination light from a single or multiple light sources. The incandescent light is more uniform than the xenon light and the intensity of both lights can be adjusted. MTF curves based on the two light sources under different intensity levels were measured and compared.

Figure 5 shows the MTF curves at the ROIs of A (Figure 5a,b) and B_2_ (Figure 5c,d) under the incandescent light (Figure 5a,c) and xenon light (Figure 5b,d), respectively. Since the incandescent light is uniform for both ROIs of A and B_2_ and the image background (same as the light portion of the edge) is around midtone level for intensity levels 3 and 4 (green numbers in Table 2), the two MTF curves at each ROI under the incandescent light at intensity levels 3 and 4 were averaged as reference MTF curves (the center and left reference curves in Figure 5).

For location A, the MTF curves for incandescent light at level 1 and xenon light at levels 1 and 2 have a larger difference from the reference curve than the MTF curves at other illumination conditions. From Table 2, the light portion intensity of the images (background) for these curves is less than 28. Frequencies where the measured MTF is below 0.1 can be susceptible to image noise, making MTF differences in these regions less critical [26].

For location B_2_, the MTF curves exhibit a larger difference from the reference curve for the incandescent light at levels 1 and 2 than at other levels, and all MTF curves show a significant difference from the reference curve for the xenon light. Again, from Table 2, the light portion intensity of the images for these curves is less than 28.

While low image luminance intensity can cause errors in MTF measurement, non-uniform image luminance can also significantly affect the measurement. To measure the image luminance uniformity, a Spectralon^®^ diffuse reflectance target was illuminated with the two light sources, and its images were captured by the endoscope, as shown in Figure 6a,b. The intensity of image luminance along the horizontal line that crosses through the image center was normalized in Figure 6c,d. The normalized intensity curves are the same for different intensity levels. From Figure 6, the largest image luminance variations with the center intensity as a reference are about 10% (Figure 6c) and 90% (Figure 6d) for the incandescent light and the xenon light respectively.

In our study, we addressed the luminance uniformity of chart images instead of the illuminance on the charts. Uniform illuminance on the chart cannot guarantee the uniform luminance of the chart image. The endoscope optic characteristics such as vignetting also affect the image luminance uniformity. For MTF calculation based on digital images, we only considered the uniformity of the final images. Therefore, the edge reflectance and contrast in the chart only approximates the intensity and contrast of image luminance.

### 3.2. $M_{chart}$ Compensation

The dashed black curves in Figure 7a,b are the actual $M_{chart}$ in the experiment, where the value at the Nyquist frequency ($f_{Nyq}$, the vertical lines) of 4.48 cycles/mm (see Section 4.8) is 0.3. The dashed and solid orange curves in Figure 7a represent the measured MTF ($MTF_{measured}$) and compensated MTF (MTF = $MTF_{measured}/M_{chart}$) results, respectively, at the center position A. The two curves almost overlap, indicating that the chart compensation does not significantly affect the MTF measurement. However, this is not the case for an endoscope with a better MTF.

We further studied the $M_{chart}$ compensation effects through simulations to cover a broader range of $M_{chart}$ and the endoscope MTF. To evaluate the impact of $M_{chart}$ compensation on endoscopes with different qualities, we artificially generated three better $MTF_{measured}$ curves (dashed green, blue, and red curves in Figure 7b) than the measured one (dashed orange curve in Figure 7a). From Figure 7b, we can see that for the same $M_{chart}$, the difference between $MTF_{measured}$ and MTF becomes larger for a better $MTF_{measured}$.

To evaluate the impact of $M_{chart}$ on MTF measurements, we recalculated the three MTF curves in Figure 7b based on two simulated $M_{chart}$ curves (the black dashed curves in Figure 7c,d) that are better than the actual one (dashed black curves in Figure 7a,b) in the experiments. The modulation values of these two $M_{chart}$ curves at the $f_{Nyq}$ are 0.7 and 0.9, respectively. A comparison of Figure 7b–d indicates that the impact of $M_{chart}$ on MTF becomes less significant when $M_{chart}$ becomes better, even for high $MTF_{measured}$ curves.

### 3.3. Linearity of Image Digital Values

The effects of linearity of image digital values were evaluated under two scenarios: low-contrast chart edge and high-contrast chart edge. The images of low-contrast edges were captured from the default chart in this study (Figure 4). The images of high-contrast edges were captured from the back-illuminated USAF 1951 chrome-on-glass chart at 40 mm. The edge contrasts of these two charts are 0.53 and 0.98, respectively, without saturation, as shown in Figure 8a.

Based on the captured edge images, we calculated the MTF curves with different encoding gamma values of 0.36, 0.45, and 0.56 and without decoding gamma correction. The MTF curves for γ = 1 represent linear data with decoding gamma correction. Figure 8 illustrates the effects of decoding gamma correction for both low- and high-contrast edges. In Figure 8b, all curves almost overlap, indicating that decoding gamma correction has no significant effects on the calculated MTF curves based on low-contrast edge images. This is consistent with the adoption of the moderate-contrast edge features in ISO 12233. In Figure 8c, however, we can see that decoding gamma correction significantly improved the MTF results for high-contrast edges. These MTF curves agree with the conclusion by Peter Burns for digital still cameras [27].

### 3.4. Auto Gain Control (AGC)

Figure 9 shows two MTF curves with AGC on and off, respectively. From the figure, AGC slightly improved the MTF in the frequency range of 0.7 to 1.3 cy/mm. However, this improvement is negligible, and within ordinary measurement variability. This indicates that the AGC has a minimal effect on the overall MTF performance of our endoscopic system under the current experimental setup. However, this conclusion cannot be extended to other application scenarios, as will be discussed in Section 4.4.

### 3.5. Image Enhancement

Edge images (Figure 10a) were captured without image enhancement and with three enhancement modes/levels: A1, A8, and B1. The edge profiles are shown in Figure 10b and the related MTF curves are shown in Figure 10c. For enhancement mode A, the A8 enhancement level is much stronger than the A1 level, resulting in a significantly higher MTF curve for A8. Both the A1 and A8 MTF curves peak higher than one. This is because the contrast enhancement boosts the contrast through higher spatial frequencies in images, making edges and details appear sharper and more pronounced. A1 and B1 have the same enhancement level but belong to different enhancement modes. Enhancement mode B enhances the contrast of even finer patterns than mode A in images. Therefore, the MTF curve for B1 is higher than the curve for A1 for frequencies higher than 1 cy/mm but lower than A1 for lower frequencies.

### 3.6. Image Compression

The MTF curves calculated from the TIFF, SHQ, HQ and SQ images captured by the endoscope are shown in Figure 11a. To further investigate the impact of image compression, the TIFF image captured by the endoscope wase compressed with MATLAB to different quality scales of JPEG images, and the MTF curves based on these images are compared in Figure 11b. The figure shows that the MTF curves for different qualities of images are similar. The curves for some JPEG images with lower quality are even slightly better than the TIFF image, probably because of the discrete cosine transform (DCT) compression (also known as block compression) artifacts. JPEG images are often obtained with the DCT compression algorithm that divides an image into square blocks (8 × 8 pixels for standard DCT) which are processed independently from each other, causing blocky compression artifacts (also called blocking artifacts). The artifacts (Figure 12) can distort the edge profile by increasing sharpness at the borders between the blocks and sometimes add a positive bias on the measured MTF, even though it obviously degrades image quality.

A high-quality edge image from an online source [24] was also compressed using the same method as described for Figure 11b. The resulting MTF curves based on these compressed images are shown in Figure 13a. The relationship between these curves is notably different from Figure 11b. In Figure 13a, the MTF curves from the compressed JPEG images are generally lower than the curve for the TIFF image. However, the figure does not exhibit a clear trend indicating that a stronger compression leads to a worse MTF curve. Since the MTF analysis is based on the edge feature in the image, this is likely due to two aspects of image compression. The spatial processing of the DCT blocks can reduce edge modulation. However, the quantization of the coefficients can introduce abrupt edge transitions that can increase the apparent edge modulation (and MTF) in the same way edge clipping can. As we can see, the measured MTF curve for the highest compressed JPEG image with a quality scalar of 20 shows an artificially higher MTF at low frequency, and the curve for the image with a quality scalar of 40 is better than the curve for the image with a quality scalar of 60.

From Figure 11 and Figure 13, it is apparent that the influence of image compression depends on the initial image’s quality and characteristics, and the compression scales are not linearly related. The content of the TIFF image for Figure 11 is shown in Figure 4, while the content of the TIFF image for Figure 13a is shown in Figure 13b.

### 3.7. ROI Dimensions

The edge-based MTF method can be used with different sizes of analysis regions (i.e., ROI dimensions). For a given ROI, the dimension perpendicular to the edge determines the spatial frequency sampling of the measured MTF. The dimension parallel to the edge influences the mitigation of the influence of image noise on the measurement. For many applications, a typical minimum analysis region is 100 × 60 pixels parallel and perpendicular to the edge.

MTF curves based on different sizes of ROI dimensions are shown in Figure 14. The figure presents results for two scenarios: (1) keeping the dimension perpendicular to the edge direction (perpendicular direction) as 60 pixels and changing the dimension parallel to the edge (parallel direction) at levels of 30, 40, 60, 80, 100, and 120 pixels (Figure 14a) and (2) keeping the parallel direction as 80 pixels and changing the perpendicular dimension at levels of 20, 40, 60, 80, and 100 pixels (Figure 14b).

From Figure 14a, the MTF curves for the parallel dimension less than 60 pixels are higher than the curves for the longer parallel dimensions. When using the slanted-edge method to evaluate the MTF of an imaging system, the MTF curves are calculated by combining all the vectors in the perpendicular direction, with the parallel dimensions as the total number of vectors. Selecting a small parallel dimension on the edge image can lead to a higher MTF curve due to several reasons. If the parallel dimension is too small, it may not capture enough edge detail, leading to aliasing, and thus introduce artificial low-frequency components in the edge spread function and inflate the calculated MTF at these frequencies. A small parallel dimension can also be more susceptible to noise, especially at higher frequencies where the signal-to-noise ratio is lower, which can artificially enhance high-frequency components in the edge spread function, leading to a higher measured MTF at high frequencies.

From Figure 14b, the MTF curves for the perpendicular dimension less than 40 pixels are higher than the curves for the longer parallel dimensions. This is probably because small perpendicular dimensions will not totally cover the edge profile. From Figure 14c, a perpendicular dimension of 20 pixels barely covers the edge profile under the ideal situation where the edge center and the ROI center overlap. Since the edge is slanted, our data show that for parallel dimensions between 60 and 120 pixels, the perpendicular dimension should be larger than 40 so that each row or column perpendicular to the edge can cover the whole edge profile, assuming the edge center and the ROI center are close.

## 4. Discussion

### 4.1. Intensity and Uniformity of Image Luminance

The ISO 12233 standard requires that the luminance of the test chart be sufficient to provide an acceptable camera output signal level [20]. The standard does not specify what level of camera output signal is adequate. The ISO 14524 standard for OECFs suggests that digital values of the background in the chart image should be close to the midtone [22]. In the zone system, Zone V that is often associated with an 18% reflectance represents a midtone [28,29]. The linear midtone digital value, i.e., the digital value before encoding gamma correction or after decoding gamma correction, for an 8-bit image is 46 (i.e., 255 × 0.18). The midtone digital value after encoding gamma correction is 118, assuming a gamma of 0.45 (i.e., $0.18^{0.45}\times 255$). The intensity and contrast requirements in the ISO 12233 standards are evolving. The 2017 version of the standard recommends that the modulation contrast of the edge be between 0.55 and 0.65 without mentioning edge intensity [30]. The 2023 version of the standard recommends that the reflectance of the dark portion of the edge should be approximately 5% [20]. Based on these standards, an ideal edge chart should have reflectance values of 18% and 5% for the light and dark portions, respectively, and contrast between 0.55 and 0.65.

Based on Figure 5 and Table 2, we recommend that the linear intensity should be between 28 and 80 for the background (light portion of the edge) and between 6 and 25 for the dark portion for an 8-bit image, which can be converted to a reflectance of between 11% and 31% for the light portion and between 2% and 10% for the dark portion. The contrast of the edge should be between 0.50 and 0.65. These ranges cover the recommendation by the standards.

The ISO 12233 standard specifies that the test chart should be uniformly illuminated, with the illuminance at any position within the chart being within ±90% of the illuminance at the chart center [20]. However, this requirement can be interpreted in two different ways. For a typical consumer camera, whose MTF is evaluated over the whole FOV, the 10% variation requirement should be applied to the whole image. However, endoscopes have a significantly different optical design from consumer cameras, causing severe geometric distortion [18]. As a result, the MTF at different ROIs within the endoscopes’ FOV varies and needs to be evaluated separately as required by the ISO 8600-5 standard. Therefore, it may be more appropriate to set the illuminance uniformity requirement at each ROI separately.

In Figure 6b,d, the image intensity at B_2_ is only 44% of the intensity at A for the xenon light, which does not satisfy the standard requirement of 10% variation within the whole chart image. However, the intensity variation within the ROI of A (80 × 60 pixels) is less than 10%, calculated as $\left(max-min\right)/max$, where max and min are the maximum and minimum pixel values within the ROI after decoding gamma correction. The variation within the ROI of B_2_ for the xenon light is 33%, which explains why all the MTF curves in Figure 5d have errors even if the intensity is at the high level of 5. For the incandescent light, the variation within the ROIs of both A and B_2_ is less than 10%.

### 4.2. Chart Modulation Compensation

The quality of the test chart is crucial in MTF measurements because it directly impacts accuracy and reliability. High-quality charts provide precise edge definitions, uniform patterns, proper contrast, and adequate resolution, all essential for accurate MTF calculation and meaningful comparisons across different systems. They should be made from durable materials that remain stable under various conditions, ensuring consistent results. Poor-quality charts can introduce errors and inconsistencies, undermining the integrity of the MTF assessment and potentially leading to flawed evaluations of optical performance.

Reports indicate that the quality of the test chart, particularly its edge modulation $M_{chart}$, has a notable impact on the consistency of MTF measurements [31]. Whether $M_{chart}$ compensation is necessary for endoscope MTF measurement depends on both $M_{chart}$ and $MTF_{measured}$. While a previous study suggests the requirements for $M_{chart}$ [31], this study did not consider the impact of $MTF_{measured}$ itself.

Figure 7 demonstrates the impacts of $M_{chart}$ compensation for four different levels of $MTF_{measured}$ curves and three different levels of $M_{chart}$ curves. From this figure, we can see that $M_{chart}$ compensation has no significant effect on MTF measurement for the following scenarios: (1) $M_{chart}$ is larger than 0.3 at $f_{Nyq}$ and $MTF_{measured}$ is less than 0.1 at 0.3 times of $f_{Nyq}$; (2) $M_{chart}$ is larger than 0.7 at $f_{Nyq}$ and $MTF_{measured}$ is less than 0.1 at 0.5 times of $f_{Nyq}$; (3) $M_{chart}$ is larger than 0.9 at $f_{Nyq}$ and $MTF_{measrued}$ is less than 0.1 at 0.7 times of $f_{Nyq}$. A summary of our findings, with a similar study as the reference [31], is shown in Table 3.

It is worth noting that $f_{Nyq}$ has units of cy/mm in the object space. For an endoscope, a decrease in the measuring distance will increase magnification and thus $f_{Nyq}$ (see Section 4.8). Since $M_{chart}$ monotonically decreases with frequencies, a chart with sufficient quality for MTF measurement at a longer distance might not be adequate at a shorter distance. In our case, the chart cannot be used to measure the MTF of our endoscope at distances shorter than 80 mm.

For a high-quality chart, $M_{chart}$ is high, resulting in minimal impact on $MTF_{measured}$. Therefore, $M_{chart}$ compensation might not be necessary, especially for an endoscope with a low MTF. For the same $M_{chart}$, compensation is more important for better $MTF_{measured}$ curves. In summary, a shorter measurement distance and a better endoscope (higher $MTF_{measured}$) have a higher requirement for chart quality, and the compensation is more important, especially if $M_{chart}$ is not high. If $M_{chart}$ is less than 0.3 at $f_{Nyq}$, the chart cannot be used for MTF measurement; even the compensation is not reliable since the error will be magnified [31].

While Table 3 provides technical guidance for $M_{chart}$ compensation, it might be burdensome to decide whether compensation is needed after analyzing $M_{chart}$ and $MTF_{measured}$. An easy way is to perform $M_{chart}$ compensation for all measurements, assuming $M_{chart}$ is known and larger than 0.3 at $f_{Nyq}$, to enhance the consistency and accuracy of MTF measurements from various endoscopes, charts and measuring distances. The “optional” recommendation in Table 3 simply means the bias will not be noticeable, i.e., the effect of $M_{chart}$ compensation is minimal. In many cases, no $M_{chart}$ compensation could lead to the underestimation of the system MTF, i.e., bias is introduced into the measurement, especially when $M_{chart}$ is not high enough but $MTF_{measured}$ is relative high as specified in Table 3. For macrophotography and endoscopic applications, where the requirements are often not satisfied, compensation is needed.

If $M_{chart}$ is unknown, this should be clearly reported together with the MTF measurement results. In this case, MTF curves have no quantitative meaning and can only be used to compare the relative performance of different endoscopes if they are measured with the same setting and method.

### 4.3. Linearity of Image Digital Values

We used a high-contrast USAF 1951 chrome-on-glass chart to evaluate the effects of linearity of image digital values on MTF measurement. This chart was used only for demonstration purposes. The current version of ISO 12233 [20] does not recommend such high-contrast edge charts to avoid any signal clipping in the digital image, which would introduce artificial edge transitions and usually inflate the measured MTF. High-contrast edge charts also place greater demands on the linearity of the endoscopic camera over a broader range of illumination levels and might trigger image enhancement algorithms. As such, the edge contrast should be modest for MTF measurements.

The 2017 version of ISO 12233 [30] required that the modulation contrast shall be between 0.55 and 0.65, but the 2023 version [20] does not specify a modulation contrast requirement. Based on our results, decoding gamma correction can be optional – i.e., the linearity of image digital values is not important – if the edge contrast is below 0.65.

As the best practice, however, we recommend decoding gamma correction to obtain linearized image digital values for MTF calculations for all scenarios for several reasons. Firstly, linearity simplifies the mathematical analysis and interpretation of the MTF. When the image digital values are linear, the MTF can be easily interpreted in terms of how the system responds to different spatial frequencies without the complications introduced by nonlinearities. Secondary, linearization helps maintain consistency in measurements. In linear systems, the response is directly proportional to the input, making it easier to compare MTF values across different systems or under various conditions. Thirdly, Fourier transform behaves linearly, and the MTF is often analyzed through Fourier transform in the frequency domain. The close relationship between linearity and the Fourier transform makes it a valuable tool for analyzing linear systems and understanding how they respond to different frequencies in the input signal. For a system that behaves linearly, the Fourier transform can be applied to each component of the system separately. If the encoding gamma is not available and thus not convenient to obtain linear image digital values through decoding gamma correction, it should be clearly specified.

### 4.4. Auto Gain Control (AGC)

We did not see a significant impact of AGC on MTF measurement in our study. However, the results cannot be extrapolated to other conditions. Changes in lighting, the test chart, image processing software, or the AGC algorithm itself could significantly affect the impact of AGC on endoscope MTF measurement. This is particularly true if the AGC step introduces the clipping of the edge feature, although this is easy to detect as part of the MTF analysis.

AGC adjusts signal gain to maintain balanced image brightness, particularly in changing lighting. The extent of this adjustment can vary, leading to different regions of the image receiving varying levels of gain based on the specific algorithm and implementation. Due to AGC, the inherent linear relationship between pixel values in the images may be compromised. Consequently, the essential linearity required for MTF measurements may no longer be preserved. Additionally, AGC can introduce certain artifacts or issues. While amplifying signals, it may also amplify noise, especially in low-light situations. In high dynamic range (HDR) scenes, AGC may diminish contrast by equalizing brightness in both dark and light areas, potentially causing the loss of detail in highlights and shadows. Rapid and frequent adjustments by AGC to changing lighting conditions can also lead to undesirable temporal artifacts. Ultimately, the impact of AGC artifacts depends on the specific implementation in the imaging device and the characteristics of the scene being captured. To mitigate these artifacts, especially in situations where precise image quality is crucial, it may be beneficial to disable AGC or use it judiciously. In applications like photography or videography where control over the imaging process is essential, manual gain control or exposure settings may be preferred.

In summary, AGC can vary gain across the image, compromising pixel value linearity for MTF measurements. It can also amplify noise, reduce contrast in HDR scenes, and cause temporal artifacts. To mitigate these issues, disabling AGC would be beneficial during MTF measurement. In cases where AGC is not controllable, this information should be explicitly mentioned in the test report.

### 4.5. Image Enhancement

The techniques of image enhancement encompass contrast and sharpness boosting, noise reduction, spatial frequency filtering, dynamic range adjustment, bilateral filtering and more. Increasing the contrast between features and enhancing edge and texture sharpness can increase the measured MTF. Noise reduction can maintain image clarity and prevent spatial frequency degradation, positively impacting the measured MTF. Spatial frequency filtering can either enhance or diminish certain details in the image depending on the specific frequencies targeted, affecting the system MTF accordingly. Adjusting dynamic range through methods like tone mapping or HDR imaging can also influence the MTF by altering contrast representation across different intensity levels in the image. Bilateral filtering can preserve edges while reducing noise.

While image enhancement in general can improve the apparent quality of images, thus increasing the MTF, excessive image enhancement, if not properly controlled, can introduce artifacts that can distort the representation of spatial frequencies and reduce the overall image quality. Image enhancement often needs to be avoided in medical imaging because it might remove details that could be medically important [21].

In summary, the relationship between image enhancement and the MTF is complex and depends on the specific enhancement methods employed. The MTF measured under image enhancement is sometimes misleading. Therefore, we recommend turning off all image enhancement functions when measuring the endoscope MTF. If the image enhancement setting cannot be controlled, it should be clearly explained in the test report, and the measured MTF should be interpreted in the context of the specific imaging system and application.

### 4.6. Image Compression

The results suggest that the impact of image compression on MTF measurement can vary depending on the image content and compression method. While image compression will generally degrade image quality, increased compression will not always reduce the measured MTF. This is because compression artifacts can distort the edge features that are used for the analysis. They can introduce a positive bias into the MTF, which is not intended to be used under these conditions. The results indicate that an MTF calculated based on a low-quality JPEG image is not reliable.

Since image compression involves spatial processing and often slightly alters the spatial information in compressed images, we can expect it to influence image resolution. While users have some control over the image compression level, different systems will have different algorithms and settings. Our testing is intended to provide results for the systems testing. Therefore, lossless images (e.g., TIFF images) are preferred for MTF calculations, especially if the purpose is to compare the results against design calculations. In many cases, lossless images are not available and endoscope compression algorithms are beyond our control; then, minimally compressed images should be used, and the image format used should be reported. If a JPEG image is used for MTF calculation, the image quality scalar should be above 90 to minimize loss (Figure 12a,b). It is also recommended to provide high-resolution digital edge images like in Figure 12 for MTF calculation to demonstrate that there are no visible artifacts at the pixel level.

### 4.7. ROI Dimensions

When using the slanted-edge method for MTF evaluation, selecting appropriate ROI dimensions is important. A small parallel dimension can lead to noise bias and variability, artificially inflating the MTF curve. Similarly, a small perpendicular dimension may not fully cover the edge profile, also leading to inaccurate MTF measurements. On the other hand, the ROI should not be too large, as endoscopes do not strictly adhere to isoplanatic behavior.

Based on our study, accurate endoscope MTF measurements can be achieved with an ROI with a parallel dimension of 60–120 pixels and a perpendicular dimension larger than 40 pixels for endoscope images with dimensions of 1280 × 1008 pixels. As a simplification, a default ROI of 80 × 60 pixels for the parallel and perpendicular dimensions, respectively, can be chosen. For other image dimensions, the ROI dimensions can be adjusted accordingly.

### 4.8. The Conversion of Frequency Units between the Image and Object Spaces

The MTF is typically evaluated in object space to characterize the system’s ability to faithfully reproduce the spatial frequencies present in the object. The unit of frequency used in this paper is cycles per mm in the object space (cy/mm). The default frequency unit based on the edge image is cycles per pixel in the image space (cy/pix). These two units can be converted from each other through the following equation,
$$f_{cy/mm,\;ob}=f_{\;cy/pix}\cdot \frac{M}{P_{mm/pix}}$$
where $P_{mm/pix}$ is the pixel pitch in mm/pixel, and *M* is the magnification at the ROI. While $P_{mm/pix}$ is often unknown to users, the value of $M/P_{mm/pix}$ can be measured by capturing images of a ruler (with high resolution and accuracy) at image center A and B_2_ at a given distance (e.g., 80 mm). The value of $M/P_{mm/pix}$ is equal to N/R in pix/mm, where R is the ruler distance, and N is the number of pixels occupied by this ruler distance on the image sensor, assuming that the image is not upsampled or downsampled in processing. We need to be cautious since the captured images might not have the same pixel dimensions as the image sensor’s native pixel dimensions since the image processing algorithm could resize the output images. In this scenario, the Nyquist frequency calculation based on the images would no longer be accurate. We measured $M/P_{mm/pix}$ at an 80 mm target distance to be 8.93 pix/mm at image center A and 7.39 pix/mm at B_2_. Therefore, the Nyquist frequency ($f_{Nyq}$) of 0.5 cy/pix in the image space can be converted to 4.46 cy/mm and 3.70 cy/mm for ROIs A and B_2_, respectively, in the object space for our endoscope at 80 mm. Please note that the value of M/$P_{mm/pix}$ is a function of target distance, which will be discussed in detail in a future paper.

### 4.9. The Control of the Image Processing Pipeline

The approach we suggest involves testing endoscopic systems under consistent settings. While we may specify test targets and illumination, imaging performance evaluation should ideally be conducted with the image processing pipeline, such as AGC, set to default settings, as these are integral parts of the product. However, it will be difficult to compare two different endoscopic systems without controlling the image processing pipeline.

Controlling the image processing pipeline ensures that variations in image quality are attributed to the optical performance of the endoscopic systems rather than differences in image processing algorithms. It also allows for a more accurate assessment of the inherent capabilities of the optical components and provides a standardized basis for comparison across different devices. Moreover, by controlling these settings, we can eliminate potential biases introduced by image processing techniques such as enhancement, leading to more reliable and reproducible results. This level of control is essential for establishing clear performance benchmarks and making informed decisions regarding the suitability of endoscopic systems for specific medical applications. Therefore, we recommend evaluating an endoscopic system under two scenarios: well-controlled image processing parameters as recommended in Section 5, and default image processing parameters during normal use.

### 4.10. Other Issues

In this study, we evaluated the MTF of our endoscope at an 80 mm chart distance primarily due to limitations in chart quality. The main goal of this study is to assess how various parameters affect consistent and accurate MTF measurement, rather than evaluating the performance of our endoscopic system. Since most endoscopes have a prime lens with a fixed focal length, the measured MTF can vary with changes in target distance, as the optical performance of endoscopes may change accordingly. Most endoscopes have a depth of field ranging from several millimeters to over 100 millimeters, with a short optimal working distance. As the target distance decreases or increases from the optimal working distance, the image may become more blurred, and the MTF will degrade. Therefore, the performance of an endoscope cannot be evaluated solely based on the MTF data at the 80 mm chart distance. The MTF at a shorter distance will be more important. To understand the performance of an endoscope, it is advisable to measure its MTF at least at the nearest, optimal, and farthest working distances within the depth of field range.

This study focuses on the MTF calculated from digital images. In most cases, the digital images will be displayed on a monitor then viewed by endoscope operators. Thus, the images examined by the operator will also be affected by the video output type and resolution, monitor quality, and viewing distance and angle, among other factors. However, compared with the monitor, the performance of the endoscopic optics and sensor is the bottleneck in the whole imaging chain. Therefore, the MTF curves based on the digital images can be approximated as the performance of the entire imaging chain. Additionally, as more endoscopes apply computer-aided diagnosis, involving algorithms and computer programs such as artificial intelligence to assist healthcare professionals in interpreting medical images for more accurate and efficient diagnosis, the digital images themselves will be the direct inputs to the computer program, and thus, their quality will directly affect the diagnosis accuracy.

The main purpose of this study is to evaluate the impact of multiple factors on MTF measurements, not on the interpretation of MTF curves. While in general, a higher MTF curve indicates better performance, two MTF curves might cross with each other, indicating an endoscope might perform better at low spatial frequencies but worse at high ones than the other endoscope. People sometimes use a single number derived from an MTF curve, such as MTF50 (the frequency where MTF is 0.5 or 50%), as a measure of image sharpness; however, it has been shown that it is not reliable sometimes [32]. Other parameters, such as the area under an MTF curve [33] or acutance [34], are also used; however, there is no study of these parameters for endoscopes. It is better to evaluate the whole MTF curve case by case, based on the endoscope applications. For example, if identifying fine tissue texture is important in an endoscopic procedure, the performance at high spatial frequencies might be more important.

## 5. Conclusions

The main objective of our study is to expand the scope of the current ISO 8600-5 standard [3] to include video endoscopes, optimizing the MTF test method to address the unique characteristics of these devices. Through our research, we have identified multiple critical factors affecting MTF measurement and provided recommendations for each, as shown in Table 4. These recommendations can be useful for revising the current endoscope resolution standard or developing a new endoscope standard to include video endoscope MTF measurements.

Our findings have significant practical implications for the medical endoscope industry and clinical practice. By enhancing the accuracy and reliability of endoscope performance assessments, these guidelines will aid device innovators in developing superior endoscopic devices and benefit patients through improved diagnostic capabilities. Our findings can also be extended to the MTF measurement of other digital imaging devices.

Future research should focus on evaluating endoscope MTFs at various working distances within the depth of field range and further exploring parameters such as the area under the MTF curve and acutance for endoscopic applications. Additionally, the impact of different video output types, monitor quality, and viewing conditions on the perceived image quality should be investigated to provide a comprehensive understanding of endoscope performance in clinical settings.

## Figures and Tables

**Figure 1 sensors-24-05075-f001:**
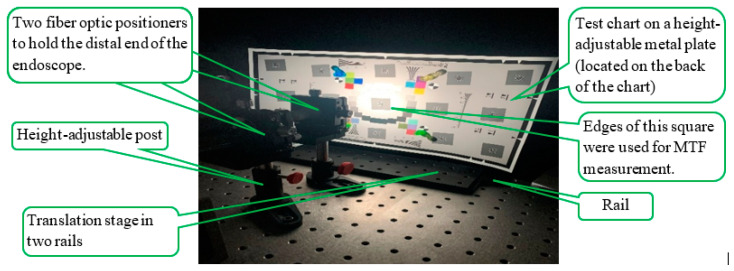
Experimental setup (using the inherent xenon light source).

**Figure 2 sensors-24-05075-f002:**
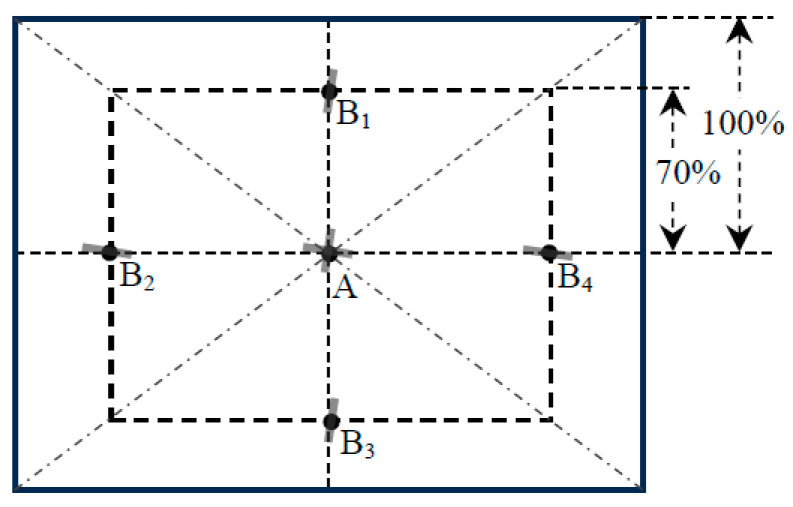
Edge center locations and directions on chart images for MTF calculation. “On-axis” point A is located at the image center. “Off-axis” points B_1_, B_2_, B_3_, and B_4_ are located at 70% of the distances from the center to the horizontal or vertical boundaries. Directions of edges: horizontal (H) at B_2_ and B_4_; vertical (V) at B_1_ and B_3_; both H and V at A (the results should be the same for square pixels).

**Figure 3 sensors-24-05075-f003:**
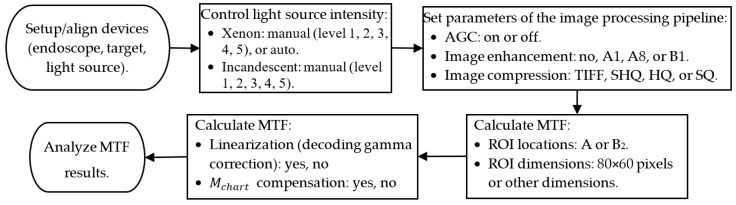
MTF measurement flowchart.

**Figure 4 sensors-24-05075-f004:**
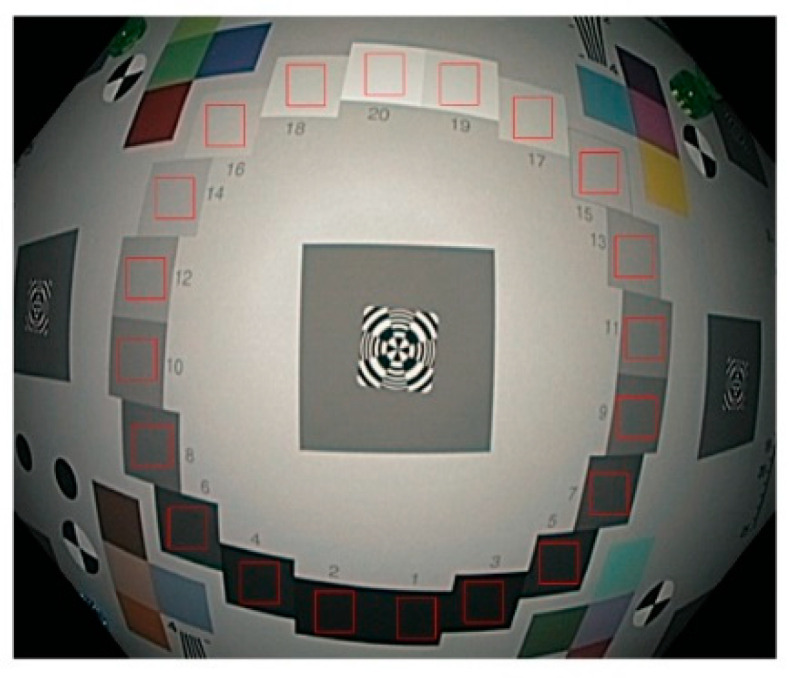
Image showing 20 gray patches on the extended ISO 12233:2017 Edge SFR chart, captured by the endoscope for OECF measurement. The average image pixel value within each red square represents the image luminance on that patch.

**Figure 5 sensors-24-05075-f005:**
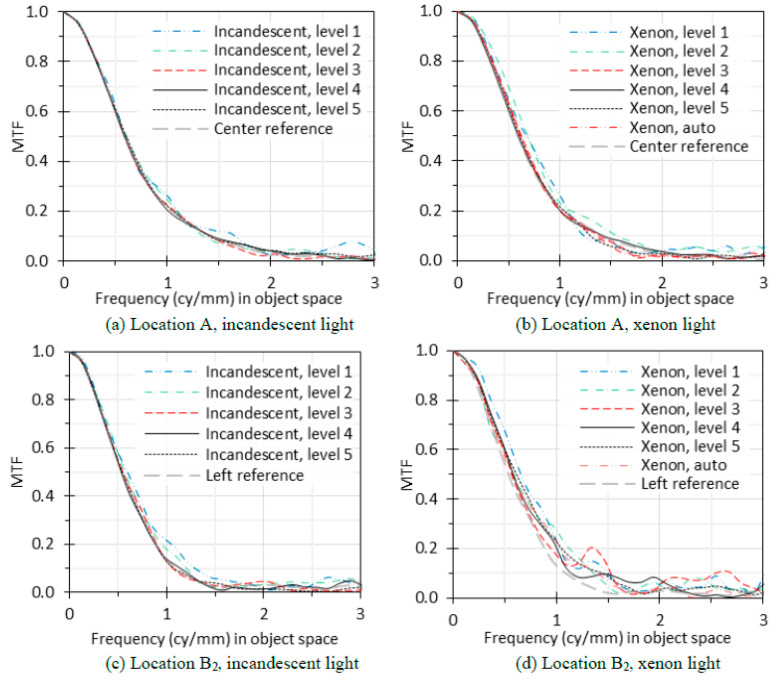
MTF curves at locations A and B_2_ under relatively uniform external incandescent light and non-uniform internal xenon light at different levels of intensities.

**Figure 6 sensors-24-05075-f006:**
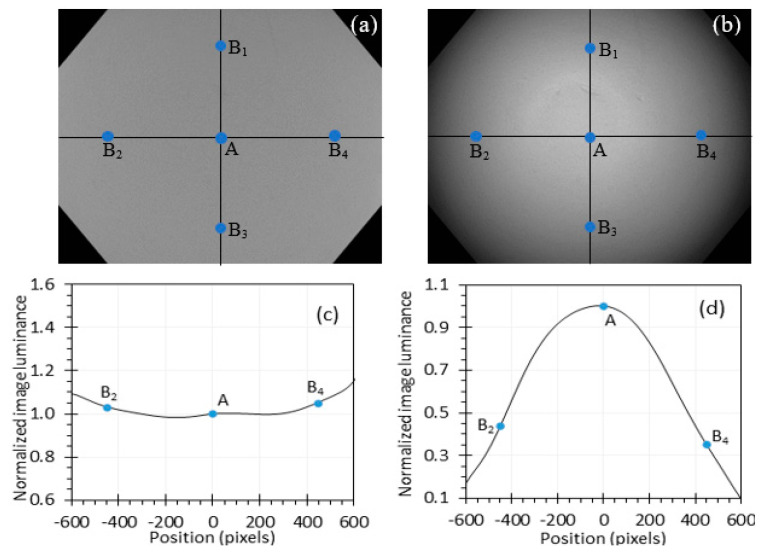
The uniformity of image luminance. (**a**,**b**): images of the Spectralon target; (**c**,**d**): normalized image luminance through the B_2_−B_4_ line, with location A as one; (**a**,**c**): incandescent light; (**b**,**d**) xenon light. The values at B_2_ in (**c**,**d**) are 1.07 and 0.44, respectively.

**Figure 7 sensors-24-05075-f007:**
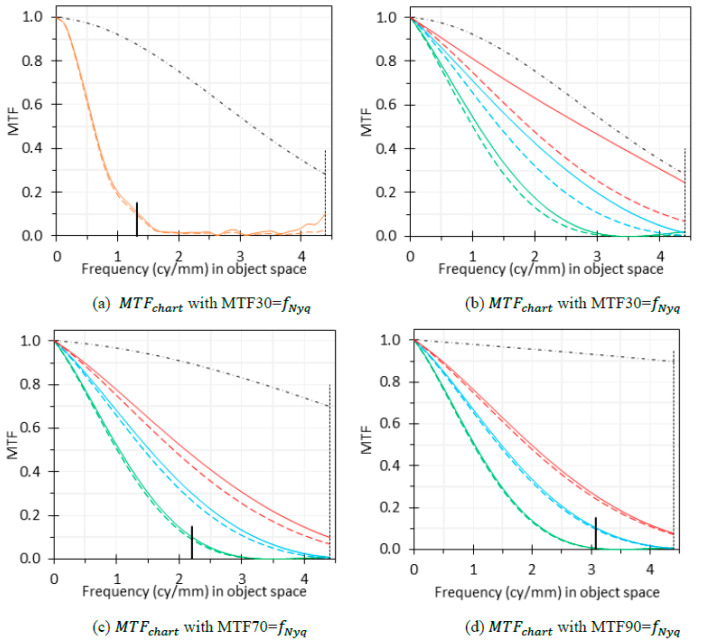
The impacts of $M_{chart}$ compensation. Colored dashed curves: the measured MTF without compensation ($MTF_{measured}$), with the orange curve in (**a**) for the center position A and the others in (**b**–**d**) artificially generated. Colored solid curves: the $M_{chart}$ compensated MTF of the dashed curves with the same color. Different colors represent different endoscope MTFs. Black dashed curves: $M_{chart}$ to calculate the compensated MTF in the same graph. Dashed vertical lines: $f_{Nyq}$. Solid vertical lines in (**a**,**c**,**d**): 0.3, 0.5, and 0.7 times of $f_{Nyq}$, respectively.

**Figure 8 sensors-24-05075-f008:**
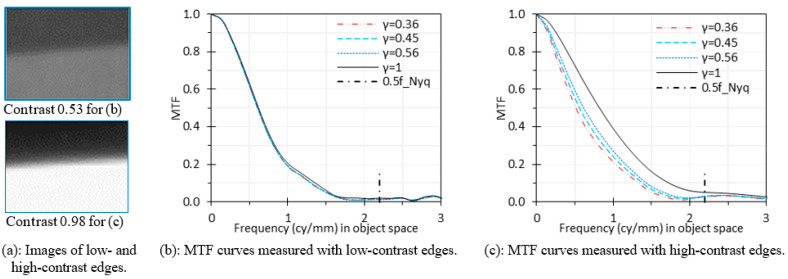
MTF curves for different encoding gamma values measured with low- and high-contrast charts. The vertical line is half Nyquist frequency.

**Figure 9 sensors-24-05075-f009:**
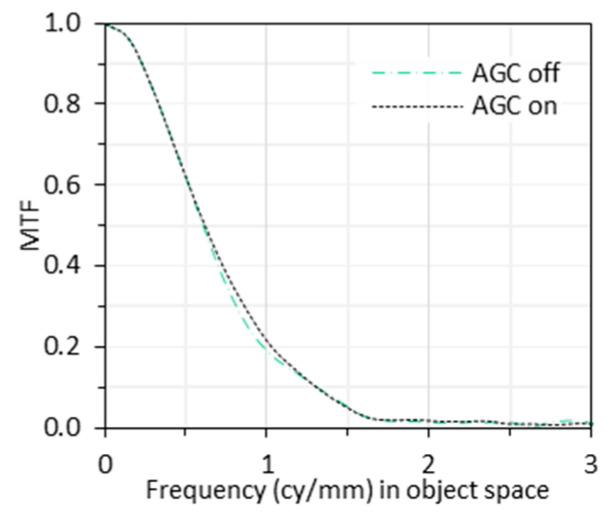
Effects of AGC on MTF curves.

**Figure 10 sensors-24-05075-f010:**
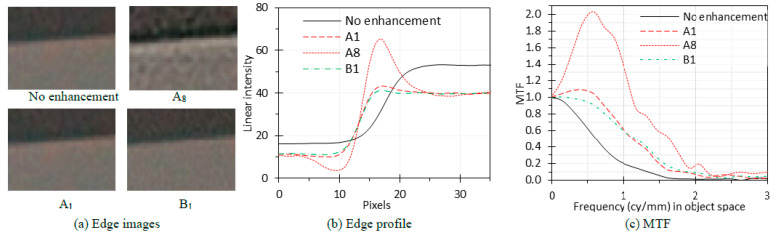
Effects of image enhancement on MTF curves.

**Figure 11 sensors-24-05075-f011:**
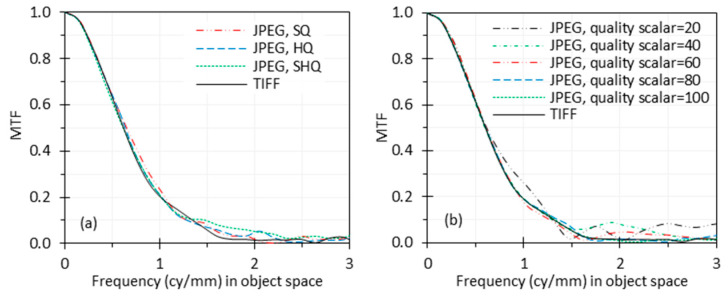
The MTF curves for an endoscope TIFF image and its compressed JPEG images (The JPEG quality scalar represents the compression level, with 100 indicating the lowest compression and highest quality). (**a**): The MTF curves calculated from the images captured by the endoscope; (**b**): The MTF curves calculated from the TIFF image captured by the endoscope and the images compressed with MATLAB from the TIFF image.

**Figure 12 sensors-24-05075-f012:**
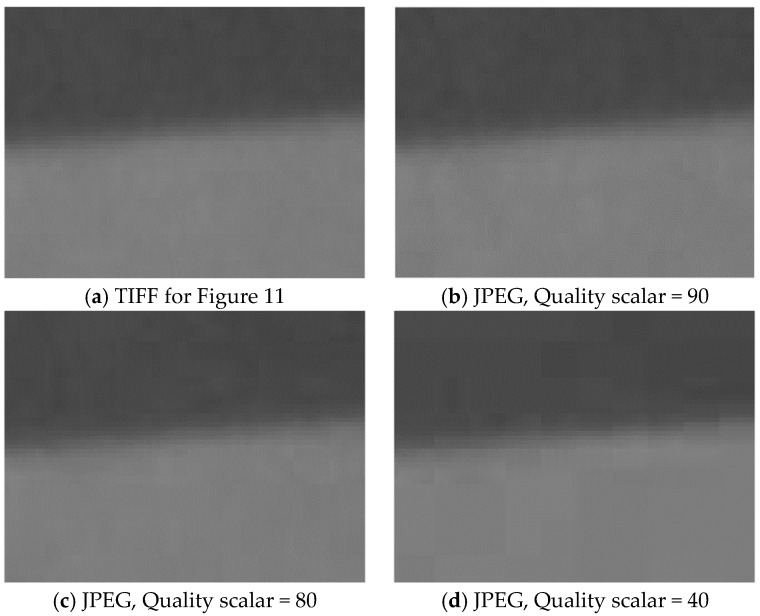
Blocky compression artifacts of edge images (80 × 60 pixels).

**Figure 13 sensors-24-05075-f013:**
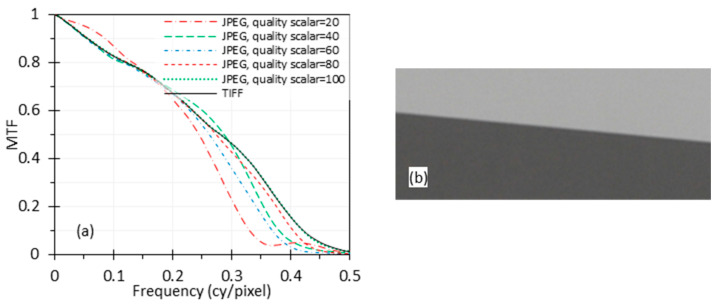
The MTF curves (**a**) for a TIFF image (**b**) from an online source [24] and its compressed JPEG images.

**Figure 14 sensors-24-05075-f014:**
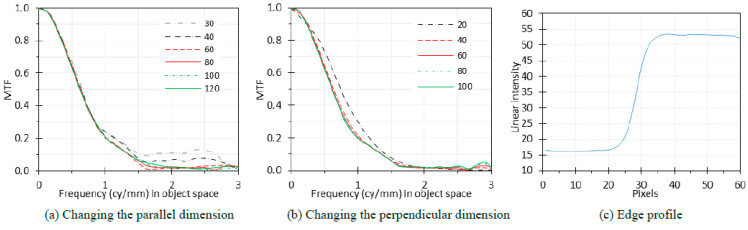
MTF curves based on different ROI dimensions (**a**,**b**) and edge profile (**c**).

**Table 1 sensors-24-05075-t001:** Default settings for studied factors.

Factors	Default Settings
Location of the ROI	On-axis, A.
Light source	Xenon light, automatic brightness control.
Chart modulation ($M_{chart}$) compensation	Measured MTF was compensated by dividing $M_{chart}$.
Linearity of image digital values	Chart images were linearized through decoding gamma correction before MTF calculation.
Auto gain control (AGC)	AGC was off.
Image enhancement	Chart images were not enhanced.
Image compression	TIFF without compression.
ROI dimensions	Parallel to the edge: 80 pixels. Perpendicular to the edge: 60 pixels.
Slanted edge directions	Horizontal at B_2_ and A.

**Table 2 sensors-24-05075-t002:** Luminance intensity and contrast of edge images under different intensity levels of xenon and incandescent lights.

		Incandescent Light					Xenon Light					
		Level 1	Level 2	Level 3	Level 4	Level 5	Level 1	Level 2	Level 3	Level 4	Level 5	Auto
	Dark portion intensity	2.0	7.0	10.9	18.9	24.1	2.2	3.8	6.6	12.1	16.3	16.3
A	Light portion intensity	9.1	30.9	42.0	64.5	79.7	10.3	17.2	28.1	42.4	52.9	52.9
	Edge contrast	0.64	0.63	0.59	0.55	0.54	0.65	0.64	0.62	0.56	0.53	0.53
	Dark portion intensity	2.7	5.2	9.2	17.4	24.7	1.0	1.3	1.9	2.9	3.8	4.0
B_2_	Light portion intensity	12.9	24.4	36.3	58.1	77.7	2.4	4.0	7.2	13.2	17.6	18.0
	Edge contrast	0.65	0.65	0.60	0.54	0.52	0.42	0.52	0.59	0.64	0.64	0.64

Note: The intensity values are 8-bit linear values after decoding gamma correction. The numbers in red are related to the MTF curves with lower accuracy in Figure 5. The numbers in green are related to the reference MTF curves. Edge contrast was computed using the linearized intensities of the light portion (I_light_) and dark portion (I_dark_) according to the formula: (I_light_ − I_dark_)/(I_light_ + I_dark_).

**Table 3 sensors-24-05075-t003:** Recommendations for chart MTF compensation.

$M_{chart}$	$MTF_{measured}$	$M_{chart}$ Compensation
>0.9 at $f_{Nyq}$	<0.1 at 0.7$f_{Nyq}$	Optional *.
	>0.1 at 0.7$f_{Nyq}$	Required.
0.7–0.9 at $f_{Nyq}$	<0.1 at 0.5$f_{Nyq}$	Optional *.
	>0.1 at 0.5$f_{Nyq}$	Required.
0.3–0.7 at $f_{Nyq}$	<0.1 at 0.3$f_{Nyq}$	Optional *.
	>0.1 at 0.3$f_{Nyq}$	Required.
<0.3 at $f_{Nyq}$	-	The chart cannot be used.

* Note: “optional” means $M_{chart}$ compensation has no significant effect on MTF measurement.

**Table 4 sensors-24-05075-t004:** Summary of factors studied: our results and recommendations for endoscope MTF measurement.

Factors Studied	Our Results	Recommendations
Intensity and uniformity of image luminance	Ensuring the proper intensity and uniformity of image luminance within the ROI is essential for accurate MTF measurements. Within the ROI, the background (light portion of the edge) intensity should be around the midtone value, the edge contrast should be within a moderate range, and the uniformity should be high.	Within the ROI, the intensity should be between 28 and 80 for the background (light portion of the edge) and between 6 and 25 for the dark portion, the edge contrast should be between 0.50 and 0.65, and the uniformity should be better than 90%. Note: All based on linearized 8-bit image luminance data.
Chart modulation compensation	Test chart quality, particularly its edge sharpness, has a notable impact on the consistency of MTF measurements, especially when $M_{chart}$ is not high and the endoscope MTF is substantial.	If $M_{chart}$ is known and larger than 0.3 at $f_{Nyq}$, divide the measured MTF ($MTF_{measured}$) by $M_{chart}$ directly to enhance the consistency and accuracy of MTF measurements. Otherwise, Table 3 can be used as a reference.
Linearity of image digital values	Linearity of image digital values can significantly affect the calculated endoscope MTF for high-contrast edges. Decoding gamma correction can be optional if the edge contrast is below 0.65.	As the best practice, performing decoding gamma correction to obtain linearized image digital values for MTF calculations in all scenarios can simplify MTF analysis and maintain measurement consistency.
Auto gain control (AGC)	While AGC did not significantly impact MTF measurements in our measurement, its effects can vary based on the system and conditions.	Disable AGC during MTF measurement to avoid potential artifacts and inconsistencies.
Image enhancement	Image enhancement techniques can significantly affect MTF measurements.	Disable image enhancement functions during MTF evaluation to ensure accuracy.
Image compression	The impact of image compression on the MTF varied with the image content and compression method and level. In some cases, compressed low-quality images yielded better MTF curves.	Lossless images (e.g., TIFF) are preferred for MTF calculations. Otherwise, minimally compressed images should be used. If a JPEG image is used, the image quality scalar should be set above 90. The format and quality of the digital edge image for MTF calculation should be reported, and the image should be provided to demonstrate the absence of visible artifacts at the pixel level.
ROI dimensions	The proper selection of ROI dimensions is crucial to avoid aliasing and noise and at the same time approximate isoplanatic behavior within the ROI.	For images of 1280 × 1008 pixels, use an ROI of 60–120 pixels for the parallel dimension and ≥40 pixel for the perpendicular dimension. An ROI of 80 × 60 pixels for the parallel and perpendicular dimensions, respectively, is recommended. Adjust the ROI dimensions accordingly for other image dimensions.

## Data Availability

The original contributions presented in the study are included in the article, further inquiries can be directed to the corresponding author.

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
