# Peer review of "Best Practices for Measuring the Modulation Transfer Function of Video Endoscopes"

_sensors, 2024, doi:10.3390/s24155075_

Round 1

Reviewer 1 Report

Comments and Suggestions for Authors

Best Practices for Measuring the Modulation Transfer Function of Video Endoscopes

Comments: 

Abstract

No comments

Introduction

    • 31-32: Replace "propelling" with "driving".

    • 37-39: Provide more detail on the importance of resolution in clinical practice.

    • 47-48: Clarify the mathematical conversion from CTF to MTF under certain assumptions in order to provide clarity on the mathematical processes or calculations used and avoid confusion.

    • 71-81: Mention specific international standards related to the measurement of MTF and OTF. 

    • 97-98: Clarify what specific intricacies and applications are mentioned in endoscopic devices.

    • 111-113: Detail why the new metrics (CTF and MTF) introduced in the 2020 standard are necessary.

    • 117-119: Detail how the exclusion of opto-electronic video endoscopes impacts the applicability of the standard.

    • 123-124:Specify which parameters affecting MTF measurement accuracy are not addressed by the current standard.

Methodology:

    • 147-148: Mention the key features of the Olympus EVIS EXERA II endoscopic system that are relevant to the study.

    • 157-158: Clarify the differences between sine-wave based graphs and edge-based graphs.

    • 169-171: Explain why a higher quality MTF measurement graph is required at a shorter distance.

    • 181-183: Justify the use of the USAF 1951 chrome-on-glass plot to assess the impact of gamma correction. 

    • 202-204: Describe in more detail the process of using the illuminance meter.

    • 223-225: Clarify the importance of measuring MTF at specific locations within an endoscopic image.

Results:

    • 428-430: Explicitly compare intensity and uniformity results between different light sources.

    • 485-487: Provide a detailed explanation of how MTF compensation graphs impact measurement accuracy.

    • 514-516: Discuss the effects of linearity and gamma correction on MTF calculations in more depth.

    • 533-535: Analyze in more detail the slight improvements in MTF when AGC is enabled.

    • 553-555: Provide a more complete comparison of MTF curves based on different image compression methods.

Discussion:

    • 637-639: Define in detail the acceptable output signal levels of the camera.

    • 677-679: Discuss in more detail the importance of test chart quality in MTF measurements.

    • 715-717: Justify the need for decode gamma correction for all scenarios.

Conclusion:

    • 884-885: Reaffirm the main objective of the study.

    • 886-888: Specifically mention how the expansion of the ISO 8600-5 standard includes video endoscopes.

    • 888-892: Summarize the identified critical factors affecting MTF measurement and recommendations for each.

    • 892-895: Briefly discuss the practical implications of these findings for the medical device industry and clinical practice.

    • 895-897: Propose directions for future research based on the limitations identified in the current study.

General

The phrase "Error! Reference source not found", appears repeatedly in the text, correct it for references to figures, tables, etc. 

Comments on the Quality of English Language

Minor typos

Reviewer 2 Report

Comments and Suggestions for Authors

1. Clearly describe how the objectives of this study differ from existing research and what specific aspects it aims to improve. It is important to clearly present the core objectives of the research and why they are significant.

2. Specify the equipment and settings used in each experiment clearly and explain them visually through diagrams or figures. For example, showing the arrangement of each device and the lighting conditions in a diagram would be helpful.

3. Specify the detailed procedures for how data was collected and analyzed step by step. It is important to describe the experimental procedures in detail so that they can be clearly understood.

4. Describe specifically how the measurement distance affects the results and why the optimal distance was chosen. For example, explain why a measurement distance of 80mm was selected as the optimal distance.

5. Discuss the limitations of the results: Address the limitations of the experimental results and propose additional research directions to address these limitations. For example, explain the limitations arising from the experimental conditions and suggest ways to overcome them.

6. Compare the MTF curves obtained using different gamma values and derive the optimal gamma value. For example, compare the MTF curves for gamma values of 0.36, 0.45, and 0.56 to find the optimal value.
